# Hermitian and Unitary Almost-Companion Matrices of Polynomials on Demand

**DOI:** 10.3390/e25020309

**Published:** 2023-02-08

**Authors:** Liubov A. Markovich, Agostino Migliore, Antonino Messina

**Affiliations:** 1Instituut-Lorentz, Universiteit Leiden, P.O. Box 9506, 2300 RA Leiden, The Netherlands; 2QuTech and Kavli Institute of Nanoscience, Delft University of Technology, 2628 CJ Delft, The Netherlands; 3Institute for Information Transmission Problems, Bol. Karetny Per. 19, 127051 Moscow, Russia; 4Russian Quantum Center, Skolkovo, 143025 Moscow, Russia; 5Department of Chemical Sciences, University of Padova, Via Marzolo 1, 35131 Padova, Italy; 6Dipartimento di Matematica ed Informatica dell’Università di Palermo, Via Archirafi 34, 90123 Palermo, Italy

**Keywords:** companion matrix, almost-companion matrix, hermitian matrix, unitary matrix, complex polynomial, density matrix, sub-parameterization

## Abstract

We introduce the concept of the almost-companion matrix (ACM) by relaxing the non-derogatory property of the standard companion matrix (CM). That is, we define an ACM as a matrix whose characteristic polynomial coincides with a given monic and generally complex polynomial. The greater flexibility inherent in the ACM concept, compared to CM, allows the construction of ACMs that have convenient matrix structures satisfying desired additional conditions, compatibly with specific properties of the polynomial coefficients. We demonstrate the construction of Hermitian and unitary ACMs starting from appropriate third-degree polynomials, with implications for their use in physical-mathematical problems, such as the parameterization of the Hamiltonian, density, or evolution matrix of a qutrit. We show that the ACM provides a means of identifying the properties of a given polynomial and finding its roots. For example, we describe the ACM-based solution of cubic complex algebraic equations without resorting to the use of the Cardano-Dal Ferro formulas. We also show the necessary and sufficient conditions on the coefficients of a polynomial for it to represent the characteristic polynomial of a unitary ACM. The presented approach can be generalized to complex polynomials of higher degrees.

## 1. Introduction

Given a complex and monic polynomial $P_{n}\left(z\right)$ (see Comment A1), it is always possible to define a matrix with a specified arrangement of the polynomial coefficients as its entries, such that $P_{n}\left(z\right)$ coincides with the characteristic polynomial of the matrix. The set $S(P_{n}\left(z\right))$ of all these n×n matrices with complex entries, sharing the same characteristic polynomial $P_{n}\left(z\right)$, is infinite and can include both derogatory and non-derogatory matrices (see Comment A2). In fact, we observe that, by definition, the *Frobenius Matrix* [1] of the shared characteristic polynomial always belongs to $S(P_{n}\left(z\right))$. A remarkable property of this matrix, which stems directly from its construction, is the coincidence between its characteristic and minimal polynomials, whatever $P_{n}\left(z\right)$. Therefore this matrix is classified as non-derogatory and, following Horn and Johnson [2], it is known as the *Companion Matrix* (CM) of its characteristic or minimal polynomial (see Comment A3).

Henceforth, we refer to the Frobenius Matrix as the Frobenius Companion matrix (FCM) of $P_{n}\left(z\right)$. When the algebraic multiplicity of each of the *n* eigenvalues of the FCM is 1, any matrix in $S(P_{n}\left(z\right))$ is non-derogatory, since these *n* distinct eigenvalues are all necessarily roots of its minimal polynomial, which therefore has degree *n* [2]. We also remark that this condition guarantees that each matrix in the $S(P_{n}\left(z\right))$ set is diagonalizable [2] and that, consequently, all matrices $\in S(P_{n}\left(z\right))$ can be generated from the FCM$\in S(P_{n}\left(z\right))$ by means of a similarity transformation. In this way, by definition and under the conditions established for the spectrum σ(FCM) of FMC, $S(P_{n}\left(z\right))$ includes all and only the companion matrices of $P_{n}\left(z\right)$.

When, instead, the distinct roots of the common characteristic polynomial are p<n, $S(P_{n}\left(z\right))$ includes infinite derogatory matrices, which cannot be structurally similar to the Frobenius matrix (see Comment A4) and to any non-derogatory matrix belonging to $S(P_{n}\left(z\right))$. For example, consider that the set $S(P_{n}\left(z\right))$ contains the diagonal matrices whose *n* entries are nothing but the *p* distinct roots of the characteristic polynomial repeated as many times as their multiplicities, whose sum is *n*. The degree of their characteristic polynomial is *n*, while the degree of their minimal polynomial is p<n [2]. Therefore, these matrices are derogatory, as are the infinite matrices generated from them by similarity.

The Frobenius matrix dates back to 1879 and was given in the form [1]:(1)$$\begin{array}{c} C_{n}=\left(\begin{array}{ccccc} 0 & 0 & \ldots & 0 & -c_{n-1}\\ 1 & 0 & \ldots & 0 & -c_{n-2}\\ 0 & 1 & \ldots & 0 & -c_{n-3}\\ \vdots & \vdots & \ddots & \vdots & \vdots \\ 0 & 0 & \ldots & 1 & -c_{1} \end{array}\right) \end{array}$$
by the German mathematician Ferdinand Georg Frobenius [3]. Sometimes, in the literature it is presented in three other unitarily transformed forms, all parameterized in terms of the *n* coefficients of the polynomial $P_{n}\left(z\right)$ and with the same number of entries equal to 0 or 1 as in (Equation 1) [4]. When the *n* eigenvalues of the Frobenius matrix are distinct, the unitary matrix that yields the diagonal form of (Equation 1) is the Vandermont matrix of its *n* eigenvalues. This property, as well as other properties and some applications of the Frobenius matrix may be found in [4].

Despite the fact that the first CM was proposed more than 140 years ago, the generation of different CMs (with further properties) has attracted a great deal of applied research activity. CMs emerge naturally in mathematical methods for finding and characterizing the roots of polynomials [5,6,7,8,9,10] and can be applied as well in the determination of solutions of high-order scalar linear differential and difference equations [11]. CMs can give matrix representations of some fields [12] and are widely used in control theory, for example in writing the controllable canonical form associated with the transfer function of a system [13]. The product of CMs is also used in the study of random walks and Markov chains [14].

The original structure of the FCM shows a modest level of flexibility and, in fact, has stimulated the search and the emergence of generalizations leading to new proposals of CMs that pave the way to applications beyond the FCM. To this end, a successful strategy is based on a radical change of the basis in which the characteristic polynomial $P_{n}\left(z\right)$ of the FCM is represented. By definition, $P_{n}\left(z\right)$ is monic and is written as the sum of $z^{n}$ and a linear combination with coefficients in C of $1,z,z^{2},\cdots ,z^{n-1}$. All these n+1 powers of *z* constitute the monomial basis, which can be replaced by other polynomial bases. In this way, one can introduce new (still non-derogatory) CMs with nonvanishing elements on the main, sub, and super diagonals. For example, the Chebyshev basis, independently adopted by Specht [15,16] and Good [17], led to a new CM called the *colleague matrix*. This approach has been further generalized by Barnett [4], who considered a basis of orthogonal polynomials and named the newly emerged CMs *comrade matrices*. Subsequently, Barnett proposed to call *confederate matrices* the CMs arising from the use of a general polynomial basis [18].

The applications dedicated to the classic problem of finding the real zeros of a real coefficient polynomial of arbitrary degree deserve a special mention, because in this context the CMs have inspired a different approach, alongside exquisitely mathematical and computational investigations [19,20,21,22,23,24]. In the last decade, new quantum theory-based root-finding algorithms exploiting the construction of Hermitian companion matrices [25,26,27,28,29,30] have also been proposed, thus increasing the interest in the study of CMs and related matrices in the rapidly growing research field of quantum computing. Furthermore, recent studies [31] have shown the opportunity of using CMs in mathematical constructions useful to the investigation of quantum entanglement, quantum state tomography, and quantum information in general.

The above is the general context for the main question addressed by this paper: is it possible to find a CM of a real or complex polynomial that is also hermitian/unitary, for example, or possesses some other prescribed special matrix structure? This question has so far been answered only partially.

In [32] the CM for a $P_{n}\left(z\right)$ with real coefficients and real zeros is constructed as a real symmetric tridiagonal Hermitian matrix. This provides a complete solution to a problem raised and partly solved by M. Fiedler in [33], which has rekindled the interest in the general structure of CMs. Note that the Frobenius matrix is itself a Fiedler matrix after a reverse permutation matrix similarity. In [34] all CMs are characterized in terms of combinatorial structure to generate new CMs. It is interesting to note that both the Frobenius and Fiedler CMs are sparse matrices, as they have 2n−1 nonzero elements [35]. A new class of sparse CMs, also known as intercyclic CMs, was introduced in [34] and includes the Fiedler matrices as a special case. In [35] the non-sparse CMs are introduced noting that they are not connected with the sparse ones by a reverse permutation matrix similarity.

To the best of our knowledge, the question of whether the CM of a complex polynomial can be sought as unitary or Hermitian is still open.

### 1.1. Purpose and Contribution of this Study

CMs generally show relatively limited versatility due to their combinatorial structure. For example, the FCM is never Hermitian or unitary. Searching for CMs that satisfy additional constraints of this kind is important when a given class of parametric polynomials is designed to reach a reliable theoretical control of a quantum physical scenario. Finding exact flexible solutions to well-defined inverse problems of this kind is a target of the present study. We stress that, for application purposes, we often do not need to associate non-derogatory matrices to a given polynomial. To emphasize this particular aspect of our matrix construction, we introduce the term *almost-companion matrix* (ACM) to refer to matrices that have a given polynomial as their characteristic polynomial but can be derogatory. Clearly, every CM is also an ACM, i.e., the set of all ACMs is a superset of the set of all CMs of a given polynomial.

In short, in this study, we address the following inverse problem: given a real or complex polynomial of any kind (for example, it may belong to a special class of polynomials, which is reflected in some special condition satisfied by its coefficients), we find a parametric ACM (see Comment A5) that satisfies preassigned conditions (e.g., those required to be Hermitian or unitary) and whose characteristic polynomial coincides with the given one. For definiteness, our investigation is here limited to complex polynomials of the third degree (n=3). The extension of the analysis to higher-degree polynomials is discussed.

The solution of the inverse problem outlined above for polynomials of order n=3 is accompanied by some useful applications. The relaxed constraints that characterize an ACM, as compared with a CM, allow the freedom to search, from the outset, for a trial ACM of a given $P_{n}\left(z\right)$ that is a Hermitian, unitary, or positive matrix, for example. If our inverse problem can be solved systematically through an approach that finds such ACMs whatever the given $P_{n}\left(z\right)$, then we readily have at our disposal a good platform for successful applications to problems such as those mentioned below.

### 1.2. Physical Applications

Constructing an ACM of a given generally complex polynomial, in addition to being interesting in itself, also has considerable applications. In elementary algebra, for example, it could be a solution tool for counting the number of the real roots (and consequently that of the complex roots in the case of a complex polynomial) of a real or complex polynomial. In addition, it may help determine a (or the only) real root of a real polynomial of odd degree. In quantum mechanics or quantum information, it could provide new parametric representations of the density operator or the evolution operator of a physical system living in a finite-dimensional Hilbert space.

The results found for a generic complex polynomial can be applied to the important particular case of a real polynomial. Investigating such a link is certainly of interest in physics. For example, in classical physics, cubic real polynomials appear when looking for the principal axes of symmetric Cartesian tensors of rank two, such as inertia or magnetic/electric dipolar tensors [36]. In quantum mechanics, they enter the scene as characteristic polynomials of any observable of a physical system that lives in a three-dimensional Hilbert space, such as a three-level atom or a qutrit. Recipes for constructing an ACM of a cubic polynomial possessing real roots after the appropriate assignment of parametric real coefficients could provide an easy way to build, e.g., Hamiltonian qutrit models on demand for control purposes, or even the density operator describing a mixed state of a three-level atom.

The paper is organized as follows. In Section 2, we formulate the inverse problem consisting of the search of the ACM for a generic complex polynomial. In Section 3, we construct the ACMs for a generic cubic complex polynomial. Through these ACMs, we introduce a way to find the roots of the given polynomial without using the Cardano-Dal Ferro formulas. The case of the polynomial with real coefficients is also discussed in detail. In Section 4 we present an application in quantum mechanics, constructing on demand the density matrix of a qutrit system as an ACM. Section 5 shows the construction on demand of the unitary ACM of a qutrit. Possible extensions to higher-degree polynomials, as well as further possible applications, are discussed in Section 6.

## 2. Formulation of the Inverse Problem

Consider a matrix $A\in M_{n}$, where $M_{n}$ is the set of all n×n matrices over the complex field C. Denoting $I_{n}\in M_{n}$ the identity matrix, the monic polynomial in the complex variable *z*
(2)$$\begin{array}{ccc} \det\left(zI_{n}-A\right) & = & z^{n}+c_{1}z^{n-1}+\cdots c_{n-1}z+c_{n} \end{array}$$
is, by definition, the characteristic polynomial of *A* and belongs to the set $P{}_{n}\left[C\right]$ of all complex monic polynomials of degree *n*.

It is well known that its *n* coefficients $c_{k}$, k=1,2⋯,n contain information about the elements of *A* that is invariant under arbitrary similarity transformations (see Comment A6). In fact, ${\left(-1\right)}^{k}c_{k}$ is the sum of all the principal minors of order *k* of *A*. In particular, $c_{1}=-\mathrm{Tr}A$ and $c_{n}={\left(-1\right)}^{n}\det A$. The profound interrelationship between a matrix and its characteristic polynomial becomes even more surprising considering the Cayley–Hamilton theorem (see Comment A7) [2] and/or Newton’s identities [37], which reveal the existence of finite algebraic expressions for the coefficients of the characteristic polynomial of a matrix in terms of traces of powers (up to *n*) of the matrix. [38,39].

The function $C:M_{n}\to P_{n}\left[C\right]$ is surjective but not injective and, hence, it cannot be inverted. In fact, it is easy to convince oneself that, for any element $P_{n}\left(z\right)\in P{}_{n}\left[C\right]$, $C^{-1}\left(P_{n}\left(z\right)\right)$ is indeed an infinite subset of $M_{n}$, since by definition it consists of all and only the matrices belonging to $M_{n}$ whose characteristic polynomial is $P_{n}\left(z\right)$.

Thus, while the direct or forward problem of finding the characteristic polynomial of a given n×n matrix is certainly well-posed according to Hadamard [40], conversely, the problem of finding a matrix $A\in M_{n}$ generating a given complex polynomial $P_{n}\left(z\right)$ is an ill-posed inverse problem [41,42], as it manifestly violates Hadamard’s uniqueness requirement, considering that every element $\in C^{-1}\left(P_{n}\left(z\right)\right)$ is a solution to the problem.

It is possible to overcome such an ill-posedness by introducing a restriction $C{|}_{{\left[C\right]}_{n}}$ of the function *C* to a subset ${\left[C\right]}_{n}$ of $M{}_{n}$, which is injectively and surjectively valued on $P{}_{n}\left[C\right]$. To this end, let us first observe that the function *C* is surjective and, by definition, $P_{n}\left(z\right)$ is the characteristic polynomial of all and only the matrices belonging to $C^{-1}\left(P_{n}\left(z\right)\right)\subset M{}_{n}$. Moreover, $C^{-1}\left(P_{n}\left(z\right)\right)\cap C^{-1}\left({P^{\prime }}_{n}\left(z\right)\right)=\emptyset$ when $P_{n}\left(z\right)\neq {P^{\prime }}_{n}\left(z\right)$. Therefore, the infinite subsets $C^{-1}\left(P_{n}\left(z\right)\right)$ of $M{}_{n}$ corresponding to the infinite *n*-degree polynomials $P_{n}\left(z\right)$ represent a partition of $M{}_{n}$. We can say equivalently that we are introducing in $M{}_{n}$ the equivalence relation A∼B consisting in the condition that $A\in M{}_{n}$ and $B\in M{}_{n}$ share the characteristic polynomial and thus belong to a given equivalence class $C^{-1}\left(P_{n}\left(z\right)\right)$. At this point, we define the subset ${\left[C\right]}_{n}$ by choosing one element from each equivalence class. According to Zermelo’s postulate, ${\left[C\right]}_{n}\neq \emptyset$ can always be constructed (in infinitely many ways in the present case, since each equivalence class is infinite), and the cardinality of its intersection with $C^{-1}\left(P_{n}\left(z\right)\right)$ is precisely one for any $P_{n}\left(z\right)$ by construction. Therefore, every *one-to-one* function $C{|}_{{\left[C\right]}_{n}}:{\left[C\right]}_{n}\to P{}_{n}\left[C\right]$ obtained by applying the axiom of choice to the quotient set $M{}_{n}/∼$ to generate ${\left[C\right]}_{n}$ is invertible. Hence, function $C{|}_{{\left[C\right]}_{n}}^{-1}:P{}_{n}\left[C\right]\to {\left[C\right]}_{n}$ defines a ${\left[C\right]}_{n}$-dependent Hadamard well-posed inverse problem, whose solution, by construction, can be given in terms of ${\left[C\right]}_{n}$ in the form
(3)$$\begin{array}{c} C{|}_{{\left[C\right]}_{n}}^{-1}\left(P_{n}\left(z\right)\right)={\left[C\right]}_{n}\cap C^{-1}\left(P_{n}\left(z\right)\right). \end{array}$$
We remark that different legitimate choices of the subset ${\left[C\right]}_{n}$ lead to different inverse problems, all well-posed in the fixed domain $P{}_{n}\left[C\right]$, and the corresponding solutions (Equation 3) differ in the generally non-similar images of one or more polynomials $P{}_{n}\left(z\right)$.

We also point out that any derogatory matrix *D* cannot be classified as a companion matrix of its characteristic polynomial $P_{D}\left(z\right)\equiv \det\left(zI_{n}-D\right)$, since *D* annihilates a polynomial having a degree lower than that of $P_{D}$ [2].

In this paper, a matrix whose characteristic polynomial coincides with a given polynomial $P_{n}\left(z\right)$ is called an *almost-companion matrix* of $P_{n}\left(z\right)$. Clearly, any CM of $P_{n}\left(z\right)$ is an ACM too. A derogatory matrix *D* such that $P_{D}\left(z\right)=P_{n}\left(z\right)$ is an ACM. In addition, a matrix similar to an ACM is still an ACM. The converse of this statement is generally false: two ACMs of the same given polynomial are not necessarily similar [43]. The set of all the ACMs of $P_{n}\left(z\right)$ cannot be generated by similarity transformations starting from an assigned ACM, since this set always includes both derogatory and non-derogatory matrices.

## 3. Almost-Companion Matrices of a Cubic Complex Polynomial

In this section, we focus on the search for an ACM of the third-degree polynomial
(4)$$\begin{array}{c} P_{3c}\left(\eta \right)=\eta^{3}+p\eta +q, \end{array}$$
which is the canonical form of
(5)$$\begin{array}{ccc} P_{3}\left(z\right) & = & z^{3}+c_{1}z^{2}+c_{2}z+c_{3}, \end{array}$$
obtained by the translation
(6)$$\begin{array}{c} \eta =z+\frac{c_{1}}{3}. \end{array}$$
The generally complex numbers *p* and *q* in (Equation 4) are related to the coefficients of $P_{3}\left(z\right)$ as follows:(7)$$\begin{array}{c} p=-\frac{c_{1}^{2}}{3}+c_{2},\;q=\frac{2c_{1}^{3}}{27}-\frac{c_{1}c_{2}}{3}+c_{3}. \end{array}$$
We denote $Q_{3c}$ the ACM of $P_{3c}\left(\eta \right)$ defined by
(8)$$\begin{array}{ccc} P_{3c}\left(\eta \right) & \equiv & \det\left(\eta I_{3}-Q_{3c}\right)=\det\left(\left(\eta -\frac{c_{1}}{3}\right)I_{3}-\left(Q_{3c}-\frac{c_{1}}{3}I_{3}\right)\right)\\ & = & \det\left(zI_{3}-\left(Q_{3c}-\frac{c_{1}}{3}I_{3}\right)\right)\equiv P_{3}\left(z\right), \end{array}$$
which means that
(9)$$\begin{array}{c} Q_{3}=Q_{3c}-\frac{c_{1}}{3}I_{3}, \end{array}$$
is the simple recipe to obtain the corresponding ACM $Q_{3}$ of $P_{3}\left(z\right)$ from $Q_{3c}$. This analysis sheds light on the advantage of first deriving $Q_{3c}$ for the simpler canonical form of a given polynomial and then finding $Q_{3c}$ from the straightforward relation (Equation 9).

Next, we formulate a trial ACM $Q_{3c}$ of (Equation 4). To this end, we observe that, in accordance with the Vieta-Girard formula for the sum of the roots of (Equation 4) [44], the absence of the quadratic term in $P_{3c}\left(\eta \right)$ implies that $tr(Q_{3c})=0$. Moreover, every matrix with elements in C is unitarily equivalent to a matrix with equal main diagonal elements [2]. Thus, it is legitimate to set the diagonal elements of our trial $Q_{3c}$ equal to zero. In constructing an ACM of (Equation 4), we aim to write its non-diagonal elements in such a way that, in the particular case of a real cubic $P_{3}$ and, hence, $P_{3c}$, the trial matrix $Q_{3c}$ becomes structurally Hermitian provided that *p* and *q* in (Equation 4) satisfy specific conditions, which will also be derived from our approach. The feasibility of this approach will highlight the greater flexibility of the ACMs compared to that of the CMs.

Following this strategy, we propose the following trial $Q_{3c}$:(10)$$\begin{array}{c} Q_{3c}\equiv Q_{3c}\left(\rho ,\varphi ,\varphi_{13}\right)=-\left(\begin{array}{ccc} 0 & \rho e^{i\frac{\varphi }{2}} & \rho e^{i\frac{\varphi }{2}}e^{i\varphi_{13}}\\ \rho e^{i\frac{\varphi }{2}} & 0 & \rho e^{i\frac{\varphi }{2}}\\ \rho e^{i\frac{\varphi }{2}}e^{-i\varphi_{13}} & \rho e^{i\frac{\varphi }{2}} & 0 \end{array}\right), \end{array}$$
where the minus sign was introduced for convenience, considering the form of the characteristic polynomial. In Equation (Equation 10), ρ is real and positive, φ is real, whereas $\varphi_{13}$ is, in general, a complex number. It is readily seen that, when φ=0 or π and $\varphi_{13}$ is real, the matrix $Q_{3c}\left(\rho ,\varphi ,\varphi_{13}\right)$ is Hermitian, consistent with our search strategy. It is useful to note that the complex conjugate of $e^{i\varphi_{13}}$ is $e^{-i\varphi_{13}^{★}}$, where $\varphi_{13}^{★}$ denotes the conjugate of $\varphi_{13}$.

The characteristic polynomial of $Q_{3c}\left(\rho ,\varphi ,\varphi_{13}\right)$ is
(11)$$\begin{array}{c} \det\left(\eta I_{3}-Q_{3c}\left(\rho ,\varphi ,\varphi_{13}\right)\right)=\eta^{3}-3\rho^{2}e^{i\varphi }\eta +2\rho^{3}e^{\frac{3}{2}i\varphi }\cos\varphi_{13}. \end{array}$$
Then, identifying polynomial (Equation 4) with (Equation 11) yields:(12)$$\begin{array}{c} \left\{\begin{array}{c} p\equiv \left|p\right|e^{i\Theta_{p}}=-3\rho^{2}e^{i\varphi }=3\rho^{2}e^{i(\varphi +\pi )},\\ q=2\rho^{3}e^{\frac{3}{2}i\varphi }\cos\varphi_{13}. \end{array}\right. \end{array}$$
Given *p*, the first Equation (Equation 12) allows us to fix ρ and select φ (in an infinite set) as follows:(13)$$\begin{array}{c} \rho =\sqrt{\frac{\left|p\right|}{3}},\;\varphi =\Theta_{p}-\pi . \end{array}$$
Defining, for p≠0, the complex parameter
(14)$$\begin{array}{c} \chi =\frac{-iqe^{-\frac{3}{2}i\Theta_{p}}}{2\sqrt{\frac{{\left|p\right|}^{3}}{27}}}, \end{array}$$
the second Equation (Equation 12) becomes an elementary trigonometric equation in C:(15)$$\begin{array}{c} \cos\varphi_{13}=\chi , \end{array}$$
which admits infinitely many solutions for any χ; in fact, similarly to the cosine function of a real variable, the complex cosine function is even and periodic with period 2π.

Using Euler’s formula, Equation (Equation 15) is easily transformed into a quadratic equation in the variable $e^{i\varphi_{13}}$, whose solution leads to
(16)$$\begin{array}{c} \varphi_{13}=-i\ln\left(\chi +{i|1-\chi^{2}|}^{\frac{1}{2}}e^{\frac{i}{2}\arg\left(1-\chi^{2}\right)}\right)=\arccos\chi . \end{array}$$
Due to the presence of the multi-valued complex function $\arg(\chi^{2}-1)$, expression (Equation 16) represents the set of infinite images of χ generated by the inverse of the non-injective cosine function over C. Therefore, strictly speaking, the expression found for $\varphi_{13}$ cannot be introduced as it is in the matrix $Q_{3c}\left(\rho ,\varphi ,\varphi_{13}\right)$. In fact, the three parameters appearing in the trial ACM (Equation 11) of (Equation 4) must be single-valued functions of the complex coefficients *p* and *q*. For our purposes, therefore, we now need to extract a specific single-valued complex function from the multi-valued function $\varphi_{13}$.

The single-valued complex function that we use here is the principal value $\Phi_{13}$ of $\varphi_{13}$, which is obtained from (Equation 16) by substituting the multi-valued functions arg and ln with their principal values, denoted Arg and Ln, respectively. This choice amounts, by definition, to constructing the principal value Arccos(χ) of function arccos(χ), which is mostly used in the literature [45,46]. It is worth noting that equation (Equation 16) can also be written in terms of $\chi^{2}-1$, but then the use of the principal value in the resulting expression for $\varphi_{13}$ would have some drawbacks, as is discussed in detail in [47].

The procedure described above gives
(17)$$\begin{array}{ccc} \Phi_{13} & \equiv & \Phi_{13}\left(\chi \right)=-i\mathrm{Ln}\left(\chi +i|1-\chi^{2}{|}^{\frac{1}{2}}e^{\frac{i}{2}\mathrm{Arg}\left(1-\chi^{2}\right)}\right)\equiv \mathrm{Arccos}\left(\chi \right)\\ & = & \mathrm{Arg}\left(\chi +i|1-\chi^{2}{|}^{\frac{1}{2}}e^{\frac{i}{2}\mathrm{Arg}\left(1-\chi^{2}\right)}\right)-iln\left(\left|\chi +i|1-\chi^{2}{|}^{\frac{1}{2}}e^{\frac{i}{2}\mathrm{Arg}\left(1-\chi^{2}\right)}\right|\right)\\ & \equiv & Re\left(\Phi_{13}\right)+iIm\left(\Phi_{13}\right), \end{array}$$
where *ln* denotes the ordinary real logarithm of its positive argument and the first term in (Equation 17) is the principal value of $\arg(\chi +{i|1-\chi^{2}|}^{\frac{1}{2}}e^{\frac{i}{2}\mathrm{Arg}\left(1-\chi^{2}\right)})$, which, by definition, generates real images in (−π,π]. Equation (Equation 17) provides the algebraic representation of the complex single-valued function Arccos(χ) whatever χ∈C. Note that, for any real χ such that |χ|≤1(|χ|≥1), the imaginary (real) component of $\Phi_{13}$ identically vanishes.

We determined all of the ingredients for constructing the trial ACM of (Equation 4) when p≠0. In accordance with (Equation 17), this ACM is a generally non-Hermitian matrix that can be written as follows:(18)$$\begin{array}{c} {\tilde{Q}}_{3c}\left(p,q\right)=\sqrt{\frac{\left|p\right|}{3}}e^{i\frac{\varphi_{p}}{2}}\left(\begin{array}{ccc} 0 & 1 & e^{i\Phi_{13}\left(\chi \right)}\\ 1 & 0 & 1\\ e^{-i\Phi_{13}\left(\chi \right)} & 1 & 0 \end{array}\right). \end{array}$$
where
(19)$$\begin{array}{c} \varphi_{p}=\Theta_{p}+\pi . \end{array}$$
Thus, based on (Equation 9), the ACM of (Equation 5) has the form
(20)$$\begin{array}{c} {\tilde{Q}}_{3}\left(p,q\right)=\left(\begin{array}{ccc} -\frac{c_{1}}{3} & \sqrt{\frac{\left|p\right|}{3}}e^{i\frac{\varphi_{p}}{2}} & \sqrt{\frac{\left|p\right|}{3}}e^{i\left[\frac{\varphi_{p}}{2}+\Phi_{13}\left(\chi \right)\right]}\\ \sqrt{\frac{\left|p\right|}{3}}e^{i\frac{\varphi_{p}}{2}} & -\frac{c_{1}}{3} & \sqrt{\frac{\left|p\right|}{3}}e^{i\frac{\varphi_{p}}{2}}\\ \sqrt{\frac{\left|p\right|}{3}}e^{i\left[\frac{\varphi_{p}}{2}-\Phi_{13}\left(\chi \right)\right]} & \sqrt{\frac{\left|p\right|}{3}}e^{i\frac{\varphi_{p}}{2}} & -\frac{c_{1}}{3} \end{array}\right). \end{array}$$
We can find an ACM of (Equation 4) for p=0 through the same kind of approach, beginning with a matrix different from (Equation 10). Our solution, denoted by ${\bar{Q}}_{3c}\left(q\right)$, can be cast as follows:(21)$$\begin{array}{c} {\bar{Q}}_{3c}\left(q\right)={\left(\frac{\left|q\right|}{\sqrt{3}}\right)}^{\frac{1}{3}}e^{\frac{i}{3}\mathrm{Arg}\left(iq\right)}\left(\begin{array}{ccc} 0 & 1 & 1\\ -e^{-i\frac{4}{3}\pi } & 0 & -1\\ -e^{i\frac{4}{3}\pi } & 1 & 0 \end{array}\right). \end{array}$$
For completeness, we also write ${\tilde{Q}}_{3c}\left(p,0\right)$(χ=0):(22)$$\begin{array}{c} {\tilde{Q}}_{3c}\left(p,0\right)=\sqrt{\frac{\left|p\right|}{3}}e^{i\frac{\varphi_{p}}{2}}\left(\begin{array}{ccc} 0 & 1 & i\\ 1 & 0 & 1\\ -i & 1 & 0 \end{array}\right), \end{array}$$
since $\Phi_{13}\left(0\right)=\pi /2$ from (Equation 17). It is not difficult to see that the eigenvalues of (Equation 22) are 0 and $\pm {\left|p\right|}^{\frac{1}{2}}e^{\frac{i\varphi_{p}}{2}}$.

Above, we formulated and solved the inverse problem of finding ACMs of generic cubic complex polynomials. Next, by exploiting these ACMs, we will present a way to find the roots of the given polynomial without resorting to the Cardano-Dal Ferro formulas, and then we will delve into the implications of having a polynomial with real coefficients on the form of the ACM and its roots.

### 3.1. Roots Characterization

Next, we assume that p≠0. Then, the eigenvalues of ${\tilde{Q}}_{3c}\left(p,q\right)$ are the roots of (Equation 4) and, hence, the roots of the characteristic polynomial of the matrix appearing in the right-hand side of (Equation 18), each multiplied by the pre-factor $\sqrt{\frac{\left|p\right|}{3}}e^{i\frac{\varphi_{p}}{2}}$. This characteristic polynomial ${\tilde{P}}_{3c}\left(\tilde{\eta }\right)$ in the unknown $\tilde{\eta }$ has the form
(23)$$\begin{array}{c} {\tilde{P}}_{3c}\left(\tilde{\eta }\right)={\tilde{\eta }}^{3}-3\tilde{\eta }-2\cos\left(\Phi_{13}\left(\chi \right)\right). \end{array}$$
The cosine representation of the free term in polynomial (Equation 23) is remarkable because it allows one to guess, at first glance, one of its three roots and then to exactly construct the other two by simply reducing the cubic polynomial ${\tilde{P}}_{3c}\left(\tilde{\eta }\right)$ to a quadratic polynomial. In fact, without resorting to the well-known Cardano-Dal Ferro formulas [48], and using instead the elementary triplication formula for the cosine function $\cos\left(3z\right)=4\cos^{3}\left(z\right)-3\cos\left(z\right)$, which also holds in the complex field, it is immediate to see that the generally complex expression (see Comment A8)
(24)$$\begin{array}{c} {\tilde{\eta }}_{1}=2\cos\left(\frac{1}{3}\Phi_{13}\left(\chi \right)\right) \end{array}$$
is a root of (Equation 23), whatever the complex coefficients p≠0 and *q*. The algebraic representation (Equation 17) of $\Phi_{13}\left(\chi \right)$ is the key to explicitly write the *p*- and *q*-dependencies of the real and imaginary components of the algebraic expression for ${\tilde{\eta }}_{1}$, which are obtained using Euler’s formula as
(25)$$\begin{array}{c} Re\left({\tilde{\eta }}_{1}\right)=2\cos\left(\frac{1}{3}Re\left(\Phi_{13}\right)\right)\cosh\left(\frac{1}{3}Im\left(\Phi_{13}\right)\right), \end{array}$$
(26)$$\begin{array}{c} Im\left({\tilde{\eta }}_{1}\right)=-2\sin\left(\frac{1}{3}Re\left(\Phi_{13}\right)\right)\sinh\left(\frac{1}{3}Im\left(\Phi_{13}\right)\right), \end{array}$$
where $Re(\Phi_{13})$ and $Im(\Phi_{13})$ are defined in (Equation 17). The other two roots are easily found to be [49]
(27)$$\begin{array}{c} {\tilde{\eta }}_{k}=-\frac{1}{2}{\tilde{\eta }}_{1}+{\left(-1\right)}^{k+1}\sqrt{3}{\left|\sin^{2}\left(\frac{1}{3}\Phi_{13}\left(\chi \right)\right)\right|}^{\frac{1}{2}}e^{\frac{i}{2}\mathrm{Arg}\sin^{2}\left(\frac{1}{3}\Phi_{13}\left(\chi \right)\right)},\;k=2,3. \end{array}$$
Equations (Equation 24) and (Equation 27) express the roots of ${\tilde{P}}_{3c}\left(\tilde{\eta }\right)$ as functions of parameter $\Phi_{13}$, which appears in the first and last anti-diagonal terms of matrices (Equation 18) and (Equation 20). We, thus, conclude that our procedure to construct the ACM also yields the roots of the (generally complex) characteristic polynomial.

It is interesting to highlight the conditions for a complex polynomial to admit real roots (note that the roots can never be all real, however, if the imaginary part of at least one of the polynomial coefficients is nonzero). To this end, it is convenient to start from the polynomial form (Equation 5). We write the three coefficients of (Equation 5) as $c_{j}\equiv x_{j}+iy_{j}$, with j=1,2,3. If a real root *r* of (Equation 5) exists, it must satisfy the equation $y_{1}r^{2}+y_{2}r+y_{3}=0$, which results from equating to zero the imaginary part of the polynomial. $\delta ={\left(y_{2}\right)}^{2}-4y_{1}y_{3}\geq 0$ is clearly a necessary condition for the existence of the root *r*, and the two only possible expressions of *r* are $\frac{-y_{2}\pm \sqrt{\delta }}{2y_{1}}$. Then, any of these two expressions is indeed a root of (Equation 5) only if it satisfies the additional condition $r^{3}+x_{1}r^{2}+x_{2}r=-x_{3}$, which results from the real part of the polynomial. We can thus state that the cubic polynomial (Equation 5) has at least one real root if and only if the inequality δ≥0 and the last condition are both met. In particular, when δ=0, *r* is a double root. Finally, we examine the case $y_{1}=0$ (and $y_{2}\neq 0$, since otherwise no real root exists for a complex polynomial). Applying the same procedure, one finds that a real root $r=-\frac{y_{3}}{y_{2}}$ of (Equation 5) exists if and only if the condition $r^{3}+x_{1}r^{2}+x_{2}r=-x_{3}$ holds. It is easy to convince oneself that this real root has multiplicity two, being a real root of the first derivative of (Equation 5).

### 3.2. Real Polynomial Case

We now investigate the special form of ${\tilde{Q}}_{3c}\left(p,q\right)$ in the case in which the three coefficients of (Equation 5) are real, with the aim of establishing properties of the polynomial roots based on its almost-companion representation built above.

In this case, (Equation 7) implies that *p* and *q* are real, and thus (Equation 4) is also a real polynomial over C. Moreover, since $\Theta_{p}$ can only be 0 (p>0) or π (p<0), the parameter χ defined in (Equation 14) is purely imaginary or real, respectively, and, therefore, its square
(28)$$\begin{array}{c} \chi^{2}=-\frac{27q^{2}e^{-3i\Theta_{p}}}{{4|p|}^{3}}=-\frac{27q^{2}}{4p^{3}}, \end{array}$$
is real for any *p*. As a consequence, using (Equation 17) it is not difficult to prove that, if and only if (see also Comment A9)
(29)$$\begin{array}{c} \Delta \left(p,q\right)\equiv \frac{p^{3}}{27}+\frac{q^{2}}{4}\leq 0, \end{array}$$
(which implies p<0, i.e., $\Theta_{p}=\pi$, and |χ|≤1), the imaginary part of $\Phi_{13}\left(\chi \right)$ given in Equation (Equation 17) vanishes, while its real part assumes the simple expression
(30)$$\begin{array}{c} \Phi_{13}\left(p,q\right)=\mathrm{Arg}\left(\chi +i|1-\chi^{2}{|}^{\frac{1}{2}}\right)=\pi -\mathrm{Arccos}\left(-\frac{3q}{2p}\sqrt{\frac{-3}{p}}\right), \end{array}$$
where we used the identity Arccos(x)=π−Arccos(−x) valid for any real *x* such that |x|≤1. It is worth noting that in the present case (Equation 27) takes the simpler form ${\tilde{\eta }}_{k}=-\frac{1}{2}{\tilde{\eta }}_{1}+{\left(-1\right)}^{k+1}\sqrt{3}\sin\left(\frac{1}{3}\Phi_{13}\left(\chi \right)\right)$ since (Equation 30) shows that $\Phi_{13}\in \left[0,\pi \right]$.

Under condition (Equation 29) and considering that $\varphi_{p}=\Theta_{p}+\pi =2\pi$, the specialization of (Equation 18) to the case under scrutiny produces the Hermitian ACM of (Equation 4) as
(31)$$\begin{array}{c} {\tilde{Q}}_{3c}\left(p,q\right)=-\sqrt{\frac{\left|p\right|}{3}}\left(\begin{array}{ccc} 0 & 1 & e^{i\Phi_{13}\left(p,q\right)}\\ 1 & 0 & 1\\ e^{-i\Phi_{13}\left(p,q\right)} & 1 & 0 \end{array}\right), \end{array}$$
where the real angle $\Phi_{13}$ is given by Equation (Equation 30). The Hermitian nature of the ACM built assures that (Equation 4), as well as (Equation 5), has three real roots. These roots are distinct when Δ(p,q)<0, while two of them are coincident if Δ(p,q)=0 [50].

We can write the real roots $x_{k}$(k=1,2,3) of (Equation 5) as follows:(32)$$\begin{array}{c} x_{k}=-\sqrt{\frac{\left|p\right|}{3}}{\tilde{\eta }}_{k}-\frac{c_{1}}{3}, \end{array}$$
where ${\tilde{\eta }}_{k}$ are the three (real) roots of (Equation 23). Equations (Equation 24) and (Equation 27), together with (Equation 30), yield
(33)$$\begin{array}{c} x_{k}=2\sqrt{\frac{\left|p\right|}{3}}\cos\left(\frac{\Phi_{13}\left(p,q\right)+\left(2k+1\right)\pi }{3}\right)-\frac{c_{1}}{3},\;k=1,2,3. \end{array}$$
It is possible to check that this formula gives the well-known trigonometric and translated forms of the three roots of (Equation 5) when they are real (see [51]).

When Δ(p,q)>0, only one of the three roots of Equation (Equation 4) or (Equation 5) (with real polynomial coefficients) is real, while the other two roots are complex conjugates. In particular, the real root corresponds to ${\tilde{\eta }}_{3}$ for p>0 and to ${\tilde{\eta }}_{1}$ for p<0. In more detail, the three roots are
(34)$$\begin{array}{cc} z_{1} & =\frac{\sqrt{p}}{2}\left(Y+iX\right)-\frac{c_{1}}{3} \end{array}$$
(35)$$\begin{array}{cc} z_{2} & =\frac{\sqrt{p}}{2}\left(Y-iX\right)-\frac{c_{1}}{3} \end{array}$$
(36)$$\begin{array}{cc} z_{3} & =-\sqrt{p}Y-\frac{c_{1}}{3} \end{array}$$
where
(37)$$X=\sqrt[{3}]{\sqrt{1-\chi^{2}}+i\chi }+\sqrt[{3}]{\sqrt{1-\chi^{2}}-i\chi }$$
and
(38)$$Y=\frac{\sqrt[{3}]{\sqrt{1-\chi^{2}}+i\chi }-\sqrt[{3}]{\sqrt{1-\chi^{2}}-i\chi }}{\sqrt{3}}$$
with
(39)$$\chi =-i\frac{3q}{2p}\sqrt{\frac{3}{p}}$$
for p>0, and
(40)$$\begin{array}{cc} z_{1} & =-\sqrt{-\frac{p}{3}}C-\frac{c_{1}}{3} \end{array}$$
(41)$$\begin{array}{cc} z_{k} & =\sqrt{-\frac{p}{3}}\left[\frac{C}{2}+i{\left(-1\right)}^{k}\sqrt{3\left(\frac{C^{2}}{4}-1\right)}\right]-\frac{c_{1}}{3},\;k=2,3, \end{array}$$
where
(42)$$C=\sqrt[{3}]{\chi +\sqrt{\chi^{2}-1}}+\sqrt[{3}]{\chi -\sqrt{\chi^{2}-1}}$$
with
(43)$$\chi =-\frac{3q}{2p}\sqrt{-\frac{3}{p}}$$
for p<0 (see derivation in Section A.1). A more elaborate derivation of the roots leading to their formulation in terms of trigonometric functions can be found in [50].

## 4. Almost-Companion Density Matrices of a Qutrit on Demand

A quantum system living in the Hilbert space H spanned by three orthonormal states |1〉, |2〉, and |3〉 is called a qutrit [52]. A pure state of the qutrit can always be represented as a normalized linear combination of these three states. To describe an arbitrary pure or mixed state of the qutrit with the same formalism, one uses instead a linear operator $\hat{\rho }$ called the density operator [53]. It acts on H and, by definition, is positive semi-definite with trace 1: tr(ρ)=1. It is well known that any positive semi-definite operator is Hermitian since its skew-Hermitian part vanishes [2,54]. As a consequence, any positive semi-definite operator is diagonalizable, and it is possible to show that its eigenvalues are real non-negative numbers. In particular, any density operator $\hat{\rho }$ is Hermitian. The three eigenvalues of the operator $\hat{\rho }$ describing the state of a qutrit are the populations of the three eigenstates of $\hat{\rho }$. The 3×3 basis-dependent matrix representation of $\hat{\rho }$, called density matrix and denoted by ρ, is also positive semi-definite and, hence, Hermitian. We observe incidentally that, conversely, any Hermitian matrix with non-negative eigenvalues is positive semi-definite and, if its trace is 1, it is a density matrix.

The purpose of this section is to demonstrate that our recipe for constructing the Hermitian ACM of a generic third-degree polynomial admitting three real roots provides an effective tool for writing density matrices of a qutrit on demand. It is worth noting that our approach itself does not require the support of a vector space, while the relationship to a basis of physical states appears in this application to a qutrit.

The aforementioned definition of density matrix results in unambiguous properties of the real coefficients of its characteristic polynomial. First of all, writing this polynomial in the canonical form
(44)$$\begin{array}{c} p_{3c}\left(\eta \right)=\eta^{3}+p\eta +q, \end{array}$$
the condition
(45)$$\begin{array}{c} p\leq -\frac{3}{2}\sqrt[{3}]{2q^{2}} \end{array}$$
stemming from Equation (Equation 29) ensures that $\Phi_{13}\left(p,q\right)$ is real, so that the ACM (Equation 31) of (Equation 44) is Hermitian and, hence, has real eigenvalues. Then, turning to form (Equation 5) of the monic polynomial through the translation (Equation 6), the Vieta–Girard formula for the sum of the roots [55] implies that the coefficient of the quadratic term is −1. Moreover, Descartes’s sign rule [56] requires that the four coefficients of polynomial (Equation 5) have alternate signs in order to have three positive roots. Therefore, the characteristic polynomial of an arbitrary density matrix of a qutrit is necessarily a third-degree real and monic polynomial of the form
(46)$$\begin{array}{c} p_{3}\left(x\right)=x^{3}-x^{2}+a^{2}x-b^{2}, \end{array}$$
where *a* and *b* are real numbers that satisfy condition (Equation 45) after translation (Equation 6). One or two roots are zero if a≠0,b=0 or a=0,b=0, respectively, while the inequality (Equation 45) is never satisfied for a=0 and b≠0.

In conclusion, under conditions (Equation 45) and (Equation 46), ${\tilde{Q}}_{3}\left(p,q\right)={\tilde{Q}}_{3c}\left(p,q\right)-\frac{c_{1}}{3}I_{3}$, with ${\tilde{Q}}_{3c}\left(p,q\right)$ given by Equation (Equation 31), is a density matrix. Incidentally, we point out that our inverse problem admits infinitely many non-Hermitian solutions, i.e., non-Hermitian ACMs of (Equation 44) or (Equation 46), such as, for example, the corresponding Frobenius companion matrix. Therefore, the explicit construction of the Hermitian ACM of (Equation 46) and, more generally, of any real third-degree polynomial with only real roots is a successful outcome of our search strategy (Equation 10). This recipe, in turn, forms the basis of the application presented below.

Let us introduce the set D of all density matrices of a qutrit in a given basis {|n〉,n=1,2,3} of H. E be the binary relation in D defined as follows: $\rho_{1}\in D$ and $\rho_{2}\in D$ are in the relation E if they are ACMs of the same polynomial $p_{3}\left(x\right)$ defined in (Equation 46). This relation, expressed by writing $\rho_{1}E\rho_{2}$, is an equivalence relation as it is manifestly reflexive, symmetric, and transitive. D is thus partitioned by E. The quotient set D/E consists of all equivalence classes of D with respect to E. Each equivalence class, which comprises all density matrices with the same characteristic polynomial, is represented by one (arbitrarily chosen) of its elements, $\bar{\rho }$, and is commonly denoted by $\left[\bar{\rho }\right]$. This is where our result (Equation 31) enters the scene, providing an easy way to parameterize the quotient set of D.

It is always possible to use the matrix ${\tilde{Q}}_{3}\left(p,q\right)={\tilde{Q}}_{3c}\left(p,q\right)-\frac{c_{1}}{3}I_{3}$ as the representative element of the equivalence class consisting of all elements of D sharing the characteristic polynomial (Equation 46), which, in turn, is uniquely associated with its canonical form (Equation 44). In this way, we establish a one-to-one correspondence between D/E and the set P of polynomials (Equation 44). This correspondence amounts to parameterizing the quotient set of D in terms of *p* and *q*. The most ambitious target of parameterizing set D is discussed in a recent topical issue [57]. It is worth emphasizing that a density matrix ρ belongs to the class of equivalence $\left[{\tilde{Q}}_{3c}\left(p,q\right)-\frac{c_{1}}{3}I_{3}\right]$ if and only if it can be unitarily generated from ${\tilde{Q}}_{3c}\left(p,q\right)-\frac{c_{1}}{3}I_{3}$, since its characteristic polynomial, trace, and Hermiticity are unitarily invariant, thus implying the invariance of the positive semi-definiteness. Therefore, while two similar matrices are ACMs of the same polynomial, a similarity transformation of a density matrix does not generate, in general, a density matrix [2].

Our parameterization of D/E in terms of the coefficients of its characteristic polynomial written in canonical form provides the theoretical basis for constructing, on demand and in a prefixed basis of H, almost-companion density matrices of any assigned polynomial $p_{3}\left(x\right)$ fulfilling the condition Δ≤0. We illustrate the concrete applicability of our recipe by constructing an almost-companion density matrix starting from the polynomial
(47)$$\begin{array}{c} p_{3}\left(x\right)=x^{3}-x^{2}+\frac{11}{36}x-\frac{1}{36}. \end{array}$$
The translation: $\eta =x-\frac{1}{3}$ yields
(48)$$\begin{array}{c} p_{3c}\left(x\right)=\eta^{3}-\frac{1}{36}\eta , \end{array}$$
so that, in this case, $p=-\frac{1}{36}$ while *q* vanishes. Exploiting Equation (Equation 30), we easily have:(49)$$\begin{array}{c} \Phi_{13}\left(-\frac{1}{36},0\right)=\pi -\mathrm{Arccos}\left(0\right)=\frac{\pi }{2}. \end{array}$$
We have, thus, obtained the few ingredients necessary to build an almost-companion density matrix of the given polynomial (Equation 47) as the sum ${\tilde{Q}}_{3}$ of the representative element of the corresponding equivalence class and the matrix $\frac{1}{3}I_{3}$, in accordance with the realization (Equation 20) of (Equation 9). That is, the density matrix has the form
(50)$$\begin{array}{c} {\tilde{Q}}_{3}=\frac{1}{6\sqrt{3}}\left(\begin{array}{ccc} 2\sqrt{3} & -1 & -i\\ -1 & 2\sqrt{3} & -1\\ i & -1 & 2\sqrt{3} \end{array}\right). \end{array}$$
Note that, while in this case it is easy to find the roots of (Equation 48) directly, and then those of (Equation 47), in general the roots of the polynomial $p_{3c}\left(\eta \right)$ corresponding to a given $p_{3}\left(x\right)$ can be found using (Equation 33).

We stress that any matrix equivalent to (Equation 50) through a unitary transformation $\hat{V}$ is an almost-companion density matrix of (Equation 47) and vice versa. For a given basis, each density matrix thus obtained describes a different (generally mixed) state of the qutrit. If, instead, the unitary transformation is interpreted as the generator of a change in the basis {|n>,n=1,2,3}, the matrix obtained represents the same density operator in the new basis $\left(\hat{V}|n>,n=1,2,3\right)$.

## 5. Unitary Matrices (Operators) on Demand

The effective construction of density matrices on demand in Section 4, results from the application of our procedure for constructing ACMs to third-degree polynomials that belong to the set P and satisfy, a priori, necessary and sufficient conditions for the existence of positive semi-definite ACMs of trace 1. By comparison, the construction of almost-companion unitary matrices on demand (that is, starting from a given appropriate third-degree polynomial) requires addressing two hurdles. The first one is to establish with certainty whether the given polynomial can be the characteristic polynomial of a unitary matrix without knowing its zeros a priori. The second difficulty lies in the fact that the trial ACM of a complex arbitrary polynomial, as given by the main Equations (Equation 9) and (Equation 10) of our procedure, is never unitary by construction. In this regard, it is important to note that the possibility of finding a non-unitary ACM of a given polynomial is not incompatible with the existence of a unitary almost-companion matrix for the given polynomial. In fact, different ACMs of a given polynomial are generally not unitarily equivalent.

In light of these considerations, we want to first identify possible structural properties shared by the coefficients of all the characteristic polynomials of a unitary matrix. Then, according to our general procedure, we will introduce a class of trial unitary matrices sufficiently representative to allow us to find a unique ACM for an assigned polynomial whose three roots are unknown but certainly have modulus 1.

### 5.1. Properties of the Characteristic Polynomial of a Unitary Matrix

It is easy to prove the following necessary and sufficient conditions concerning the characteristic polynomial of a unitary ACM:

**Theorem** **1.**
*Let $D_{m}\left(z\right)$ be any complex polynomial of degree m, with 1≤m≤n, dividing an arbitrarily given complex polynomial $P_{n}\left(z\right)$. Then $P_{n}\left(z\right)$ admits a unitary ACM if and only if any $D_{m}\left(z\right)$ does.*


**Proof.** Necessity: if $P_{n}\left(z\right)$ admits a unitary ACM, then all its roots have modulus one. This property is obviously transferred to each $D_{m}\left(z\right)$ dividing $P_{n}\left(z\right)$, which, in turn, implies the existence of a diagonal unitary ACM of $D_{m}\left(z\right)$.Sufficiency: Since $P_{n}\left(z\right)$ can be represented as the product of *n* monic binomials whose free terms are complex numbers of modulus one by hypothesis, then a diagonal ACM of $P_{n}\left(z\right)$ with entries having modulus one exists. This ACM is unitary [2]. □

When n=2, it is easy to convince oneself that

**Theorem** **2.**
*A monic complex, second-degree polynomial is the characteristic polynomial of a unitary matrix of order 2, if and only if it has the structure*

(51)
$$\begin{array}{c} P_{2}\left(z\right)=z^{2}-r_{2}e^{i\vartheta }z+e^{2i\vartheta }, \end{array}$$

*with $r_{2}\in \left[0,2\right]$ and ϑ∈(−π,π].*


**Proof.** To demonstrate this double statement it is sufficient to explicitly find the two roots of (Equation 51) for the necessity and to use simple geometric arguments (or exploit Theorem 1) for the sufficiency. □

We additionally remark that, for $r_{2}>2$, the principal arguments of the two roots of (Equation 51) coincide with χ, and the product of their modules, both different from unity, is still one.

When the order of the unitary matrix is greater than 2, it is still possible to find peculiar properties possessed by the coefficients of the corresponding characteristic polynomial. However, there are polynomials of degrees higher than 2 and structures similar to (Equation 51), which also have roots with moduli different from 1. We prove here the following useful necessary condition on the structure of the characteristic polynomial of a 3×3 unitary matrix

**Theorem** **3.**
*The complex third-degree characteristic polynomial of any unitary matrix of order 3 has necessarily the structure:*

(52)
$$\begin{array}{c} P_{3}\left(z\right)=z^{3}-re^{i\theta_{1}}z^{2}+re^{i(\theta -\theta_{1})}z-e^{i\theta }, \end{array}$$

*where r∈[0,3], $\theta_{1}\in \left(-\pi ,\pi \right]$ and θ∈(−π,π].*


**Proof.** Given any three real numbers α, β, and γ, it is always possible to find three real numbers *r*, $\theta_{1}$, and θ that satisfy the following relations:
(53)$$\begin{array}{c} e^{i\alpha }+e^{i\beta }+e^{i\gamma }=re^{i\theta_{1}}, \end{array}$$
(54)$$\begin{array}{c} e^{i\alpha }e^{i\beta }e^{i\gamma }=e^{i\theta }. \end{array}$$
The product of (Equation 54) and the complex conjugate of (Equation 53) gives
(55)$$\begin{array}{c} re^{i(\theta -\theta_{1})}=\left(e^{-i\alpha }+e^{-i\beta }+e^{-i\gamma }\right)e^{i\alpha }e^{i\beta }e^{i\gamma }=e^{i(\alpha +\beta )}+e^{i(\beta +\gamma )}+e^{i(\alpha +\gamma )}, \end{array}$$
where r∈[0,3], $\theta_{1}\in \left(-\pi ,\pi \right]$ and $\theta =\mathrm{Arg}e^{i(\alpha +\beta +\gamma )}\in \left(-\pi ,\pi \right]$. Equations (Equation 53)–(Equation 55) represent the Vieta–Girard formulas for the three roots $e^{i\alpha }$, $e^{i\beta }$, and $e^{i\gamma }$ of polynomial (Equation 52). Since these roots have modulus 1, as is required for $P_{3}\left(z\right)$ to be the characteristic polynomial of a unitary matrix, the Vieta–Girard formulas (Equation 53) and (Equation 54) clearly show that the complex coefficients of the characteristic polynomial of any 3×3 unitary matrix are not independent. In fact, the free term and the coefficient of $z^{2}$, which are involved in Equations (Equation 53) and (Equation 54), respectively, univocally determine the coefficient of *z* through (Equation 55), in accordance with (Equation 52). □

Similar to Theorem 1, Theorem 3 can be extended to a generic degree *n*. We emphasize that the polynomial form (Equation 52) and its roots have some remarkable properties. For example, the passage from *z* to the auxiliary variable $u=ze^{i\psi }$ leads, up to a global phase factor, to a polynomial with the same structure (Equation 52) after the angle shifts $\theta_{1}^{\prime }=\theta_{1}+\psi$ and $\theta^{\prime }=\theta +3\psi$ (these shifts are unimportant for what concerns the polynomial structure since the angles $\theta_{1}^{\prime }$ and $\theta^{\prime }$ can take the same range of values as $\theta_{1}^{\prime }$ and $\theta^{\prime }$). Therefore, the three roots of the new polynomial have the same modules and relative principal arguments as the roots of the original polynomial (Equation 52). Another interesting property resulting from Equation (Equation 54) is that, if (Equation 52) admits one root with modulus one, the other two roots must have reciprocal modules (including the case in which they also have modulus one).

Note that Theorem 3 only expresses a necessary condition and, therefore, there exist polynomials with structure (Equation 52) that do not admit unitary ACM. Algebraic relations among the three parameters in the expression of $P_{3}\left(z\right)$ not implied by the structure of the polynomial itself can ensure that $P_{3}\left(z\right)$ admits a unitary ACM (vide infra).

Consider, for example, the case r=3. The polynomial $z^{3}-3z^{2}+3e^{i\pi }z-e^{i\pi }$ has the form (Equation 52), from which it is obtained by (arbitrarily) choosing $\theta_{1}=0$ and θ=π. This is not the characteristic polynomial of a unitary 3×3 matrix, since its roots are −1 and $2\pm \sqrt{3}$; accordingly, $\theta =\pi \neq \mathrm{Arg}e^{i(3\theta_{1})}=0$. The relation $\theta =3\theta_{1}$ guarantees, instead, the existence of three roots of modulus 1 when r=3, as is easily seen geometrically or from the fact that in this case $P_{3}\left(z\right)={\left(z-e^{i\theta_{1}}\right)}^{3}$. As another example, for r=1, (Equation 52) admits a unitary ACM for any $\theta_{1}\in \left(-\pi ,\pi \right]$ and θ∈(−π,π], as the roots of the polynomial are $e^{i\theta_{1}}$ and $e^{i\frac{\theta -\theta_{1}\pm \pi }{2}}$.

The analysis in the next section will provide expressions for the coefficients of polynomial (Equation 52), making it the characteristic polynomial of a unitary ACM.

### 5.2. Construction of a Trial Unitary ACM

In Section 3, it was convenient to search for an ACM of a generic monic complex polynomial (Equation 5) of the third degree in the unknown *z* by resorting to the canonical form (Equation 4) of the polynomial through a translation of the complex variable *z*. There are two advantages to using polynomial (Equation 4) in the translated variable η: the number of parameters appearing in the polynomial expression is reduced from 3 to 2 (namely, *p* and *q* instead of $c_{1}$,$c_{2}$, and $c_{3}$), and the very simple recipe (Equation 9) allows one to obtain the ACM of the given polynomial from the ACM of its canonical form (Equation 4).

It is clearly possible to pass from $P_{3}\left(z\right)$ to its canonical form through the appropriate translation of *z*. Unfortunately, such a strategy is not convenient in this case, since the canonical polynomial generally does not admit a unitary ACM, and thus the further mathematical step complicates the achievement of our goal. Therefore, we propose a different approach that combines geometrical and analytical considerations.

Exploiting Theorems 1 and 2, we can represent each element $P_{3}\left(z\right)$ of the set $\left[P_{3}\left(z\right)\right]$ of all and only the third-degree polynomials that admit a unitary ACM and share the root 1 as follows:(56)$$\begin{array}{c} P_{3}\left(z\right)=\left(z^{2}-r_{2}e^{i\vartheta }z+e^{2i\vartheta }\right)\left(z-1\right)=z^{3}-\left(1+r_{2}e^{i\vartheta }\right)z^{2}+\left(1+r_{2}e^{-i\vartheta }\right)e^{2i\vartheta }z-e^{2i\vartheta }, \end{array}$$
where $r_{2}\in \left[0,2\right]$ and ϑ∈(−π,π]. Each polynomial (Equation 56) possesses, by construction, a unitary ACM and, vice versa, the characteristic polynomial of any 3×3 unitary matrix with a unit eigenvalue is a particular realization of (Equation 56).

$\left[P_{3}\left(z\right)\right]$ is a subset of the set $\left[P_{3}\left(z\right)\right]$ of the polynomials of the form (Equation 52). This point is appreciated by noting that the coefficient of $z^{2}$ in Equation (Equation 52) can always be represented as
(57)$$\begin{array}{c} re^{i\theta_{1}}=\left(1+r_{2}e^{i\vartheta }\right) \end{array}$$
with
(58)$$\begin{array}{c} {r_{2}}^{2}=1+r^{2}-2r\cos\theta_{1} \end{array}$$
and
(59)$$\begin{array}{c} \vartheta =\mathrm{Arg}\left(-1+re^{i\theta_{1}}\right)=2\mathrm{Arctan}\left(\frac{r\sin\theta_{1}}{-1+r\cos\theta_{1}+\sqrt{1+r^{2}-2r\cos\theta_{1}}}\right). \end{array}$$
The last equality is based on the following identity [58], which gives the principal argument of a generic complex number (x+iy)∈Ω, where Ω coincides with the complex plane cut along the negative *x*-axis:(60)$$\begin{array}{c} \mathrm{Arg}\left(x+iy\right)=2\mathrm{Arctan}\left(\frac{y}{x+\sqrt{x^{2}+y^{2}}}\right). \end{array}$$
As expected, this formula leaves the argument of a complex number of null modulus undefined and, for any fixed negative *x*, implies
(61)$$\begin{array}{c} \lim_{y⟶0^{\pm }}\mathrm{Arg}\left(x+iy\right)=\pm \pi . \end{array}$$
The above equations clearly show that $\left[P_{3}\left(z\right)\right]$ is obtained as a subset of $\left[P_{3}\left(z\right)\right]$ by introducing the relations (Equation 57)–(Equation 59) among the parameters *r*, $\theta_{1}$, and θ, which are arbitrary in their ranges of definition in the polynomial expression (Equation 52). In particular, θ is compatible with (Equation 56) only if
(62)$$\begin{array}{c} \theta =2\vartheta , \end{array}$$
thus leading to the following:

**Theorem** **4.**
*A monic third-degree polynomial (Equation 52) belongs to the set $\left[P_{3}\left(z\right)\right]$ if and only if it can be written in the form*

(63)
$$\begin{array}{c} {\tilde{P}}_{3}\left(z\right)=z^{3}-\left(1+r_{2}e^{i\frac{\theta }{2}}\right)z^{2}+\left(1+r_{2}e^{-i\frac{\theta }{2}}\right)e^{i\theta }z-e^{i\theta }, \end{array}$$

*where $r_{2}\in \left[0,2\right]$ and θ∈(−π,π].*


Note that the range of $r_{2}$ values is dictated by Theorem 2, and the corresponding range of *r* values resulting from Equation (Equation 58) for any given $\theta_{1}$ is a subset of the interval [0,3] in Equation (Equation 52), in accordance with the fact that $\left[P_{3}\left(z\right)\right]$ is a subset of $\left[P_{3}\left(z\right)\right]$. On the other hand, since ϑ∈(−π,π], Equation (Equation 62) implies that θ∈(−2π,2π], which can be clearly reduced to the principal interval [−π,π]. We emphasize that requiring (Equation 52) to be the characteristic polynomial of a unitary matrix with a real positive eigenvalue (that is, 1) entailed relations between the three parameters in Equation (Equation 52), thus leading to the dependence of polynomial (Equation 63) on only two parameters, $r_{2}$ and θ.

At this point, we can construct an ACM for a polynomial of the kind (Equation 63). The polynomial factorization in Equation (Equation 56) enables a block diagonal form for the ACM, with the one-dimensional block simply equal to 1. The diagonal elements of the 2×2 block can be set equal [2] and are immediately obtained from Vieta’s formula for the sum of the roots of polynomial (Equation 51). Then, simple algebraic considerations lead to the following ACM of polynomial (Equation 63):(64)$${\tilde{W}}_{3}=\left(\begin{array}{ccc} \frac{r_{2}}{2}e^{i\frac{\theta }{2}}\; & \sqrt{1-{\left(\frac{r_{2}}{2}\right)}^{2}}e^{i\frac{\theta }{2}}\; & 0\\ -\sqrt{1-{\left(\frac{r_{2}}{2}\right)}^{2}}e^{i\frac{\theta }{2}}\; & \frac{r_{2}}{2}e^{i\frac{\theta }{2}}\; & 0\\ 0 & 0 & 1 \end{array}\right).$$
It is easy to verify that the characteristic polynomial of (Equation 64) coincides with (Equation 63) for all the allowed values of the parameters $r_{2}$ and θ. The three columns of ${\tilde{W}}_{3}$ are normalized and mutually orthogonal, and these properties imply the unitarity of matrix ${\tilde{W}}_{3}$. If $r_{2}>2$, and thus it is out of the range given in Theorem 4, the first two columns of the matrix are not orthogonal for any θ, and then ${\tilde{W}}_{3}$ is no longer unitary.

It is worth noting that (Equation 64) can be seen as the result of a partial diagonalization of another unitary matrix with the same eigenvalues and that all the other ACMs of a polynomial (Equation 63) can be obtained by unitary transformation of (Equation 64).

Once an ACM is constructed for any polynomial (Equation 63), the subset of $\left[P_{3}\left(z\right)\right]$ that contains all and only the polynomials (Equation 52) admitting a unitary ACM can be generated by rotating the roots of each polynomial (Equation 63) by an angle ϵ∈(−π,π]. This amounts to changing the complex variable *z* to $u=ze^{i\epsilon }$ in ${\tilde{P}}_{3}\left(z\right)$. Then, up to a global phase factor $e^{3i\epsilon }$, the polynomial ${\tilde{P}}_{3}\left(u\right)={\tilde{P}}_{3}\left(ze^{i\epsilon }\right)$ is equal to
(65)$$\begin{array}{c} P_{3\epsilon }\left(z\right)=z^{3}-\left(1+r_{2}e^{i\frac{\theta }{2}}\right)e^{-i\epsilon }z^{2}+\left(1+r_{2}e^{-i\frac{\theta }{2}}\right)e^{i\theta }e^{-2i\epsilon }z-e^{i(\theta -3\epsilon )}. \end{array}$$
The root 1 of ${\tilde{P}}_{3}\left(z\right)$ corresponds to a general complex root of modulus one, $e^{i\epsilon }$, of $P_{3\epsilon }\left(z\right)$. $\left[P_{3\epsilon }\left(z\right)\right]$ includes all and only the polynomials $P_{3\epsilon }\left(z\right)$, which, by construction, admit a unitary ACM and, therefore, also satisfy the necessary condition expressed by Theorem 4. We have thus proved that

**Theorem** **5.**
*A complex monic polynomial of the third degree is the characteristic polynomial of a unitary matrix of order 3 if and only if it has the structure (Equation 65), with $r_{2}\in \left[0,2\right]$, θ∈(−π,π], and ϵ∈(−π,π].*


Next, we accomplish the main objective of this section by constructing an ACM $W_{3}$ of $P_{3\epsilon }\left(z\right)$. To this end, in analogy with recipe (Equation 9), we use the transformation $z=ue^{-i\epsilon }$ to generate $W_{3}$ from the matrix ${\tilde{W}}_{3}$ in Equation (Equation 9) by proceeding as follows:(66)$$\begin{array}{ccc} {\tilde{P}}_{3}\left(u\right) & \equiv & det\left(uI_{3}-{\tilde{W}}_{3}\right)=det\left(e^{i\epsilon }\left(zI_{3}-e^{-i\epsilon }{\tilde{W}}_{3}\right)\right)\\ & = & e^{3i\epsilon }det\left(zI_{3}-W_{3}\right)\equiv e^{3i\epsilon }P_{3\epsilon }\left(z\right), \end{array}$$
where $W_{3}\equiv e^{-i\epsilon }{\tilde{W}}_{3}$. In light of our previous arguments, the characteristic polynomial of the matrix
(67)$$W_{3}=e^{-i\epsilon }{\tilde{W}}_{3}=\left(\begin{array}{ccc} \frac{r_{2}}{2}e^{i(\frac{\theta }{2}-\epsilon )}\; & \sqrt{1-{\left(\frac{r_{2}}{2}\right)}^{2}}e^{i(\frac{\theta }{2}-\epsilon )}\; & 0\\ -\sqrt{1-{\left(\frac{r_{2}}{2}\right)}^{2}}e^{i(\frac{\theta }{2}-\epsilon )}\; & \frac{r_{2}}{2}e^{i(\frac{\theta }{2}-\epsilon )}\; & 0\\ 0 & 0 & e^{-i\epsilon } \end{array}\right),$$
is polynomial (Equation 65). The unitary matrix (Equation 67) fully meets our goal. To reach this objective, we first characterized the class $\left[P_{3\epsilon }\left(z\right)\right]$ of all and only the polynomials that admit a unitary ACM, thus removing the difficulties related to the lack of sufficiency of polynomial (Equation 52). Then, exploiting recipe (Equation 66), we established the form $W_{3}$ of a unitary ACM for any polynomial belonging to $\left[P_{3\epsilon }\left(z\right)\right]$.

In the following, we illustrate our approach through some applications.

### 5.3. Examples

Given the parameter σ=±1, consider the subclass of polynomials (Equation 65) with $\epsilon =\frac{(1-\sigma )\pi }{2}$:
(68)$$\begin{array}{ccc} P_{3\sigma^{\prime }}\left(z\right) & = & z^{3}-\left(1+r_{2}e^{i\frac{\theta }{2}}\right)e^{-i\frac{(1-\sigma )\pi }{2}}z^{2}+\left(1+r_{2}e^{-i\frac{\theta }{2}}\right)e^{i\theta }z-e^{i(\theta -\frac{(1-\sigma )\pi }{2})}\\ & = & z^{3}-\left(1+r_{2}e^{i\frac{\theta }{2}}\right)\sigma z^{2}+\left(1+r_{2}e^{-i\frac{\theta }{2}}\right)e^{i\theta }z-\sigma e^{i\theta }, \end{array}$$In Equation (Equation 68) we exploited the identity $e^{-i\frac{(1-\sigma )\pi }{2}}=\sigma$. It is easy to verify that $P_{3\sigma }\left(\sigma \right)=0$, which means that (Equation 68) is the most general polynomial with the real root σ and an ACM which, in view of (Equation 67), can be written on demand as
(69)$$\begin{array}{c} W_{3\sigma }=\left(\begin{array}{ccc} \sigma \frac{r_{2}}{2}e^{i\frac{\theta }{2}}\; & \sigma \sqrt{1-{\left(\frac{r_{2}}{2}\right)}^{2}}e^{i\frac{\theta }{2}}\; & 0\\ -\sigma \sqrt{1-{\left(\frac{r_{2}}{2}\right)}^{2}}e^{i\frac{\theta }{2}}\; & \sigma \frac{r_{2}}{2}e^{i\frac{\theta }{2}}\; & 0\\ 0 & 0 & \sigma \end{array}\right). \end{array}$$Consider the polynomial
(70)$$\begin{array}{ccc} P_{3,r_{2}=0}\left(z\right) & = & z^{3}-e^{i\theta_{1}}z^{2}+e^{i\theta^{\prime }}e^{-i\theta_{1}}z-e^{i\theta^{\prime }}, \end{array}$$
obtained by setting $r_{2}=0$, $\epsilon =-\theta_{1}$, and $\theta =\theta^{\prime }-3\theta_{1}$ in Equation (Equation 65). With these choices, using (Equation 67) we immediately find that polynomial (Equation 70) admits the ACM
(71)$$\begin{array}{c} W_{r_{2}=0}=\left(\begin{array}{ccc} 0\; & e^{i\frac{\theta^{\prime }-\theta_{1}}{2}}\; & 0\\ -e^{i\frac{\theta^{\prime }-\theta_{1}}{2}}\; & 0\; & 0\\ 0 & 0 & e^{i\theta_{1}} \end{array}\right). \end{array}$$When r=1, the polynomial (Equation 70) coincides with polynomial (Equation 52) up to a trivial change of notation (θ is substituted with $\theta^{\prime }$). This polynomial is then the characteristic polynomial of a unitary matrix for any $\theta^{\prime }$ and $\theta_{1}$, as we observed in Section 5.1, with eigenvalues that are now immediately derived from the matrix (Equation 71) as $e^{i\theta_{1}}$ and $e^{i\frac{\theta^{\prime }-\theta_{1}\pm \pi }{2}}$.For r=0, (Equation 52) yields the polynomial $z^{3}-e^{i\theta }=0$ and, whatever the ϵ value, Equations (Equation 57) and (Equation 62) imply that $r_{2}=1$, and θ=2π in Equation (Equation 65). The corresponding ACM, with structure (Equation 67), takes the form
(72)$$\begin{array}{c} W_{r=0}=\left(\begin{array}{ccc} \frac{1}{2}e^{i(\pi -\epsilon )}\; & \sqrt{\frac{3}{4}}e^{i(\pi -\epsilon )}\; & 0\\ -\sqrt{\frac{3}{4}}e^{i(\pi -\epsilon )}\; & \frac{1}{2}e^{i(\pi -\epsilon )}\; & 0\\ 0 & 0 & e^{-i\epsilon } \end{array}\right), \end{array}$$Apart from the cases r≠1 or r≠0 examined above, for a generic *r* Equation (Equation 52) admits a unitary ACM only under *r*-dependent algebraic constraints on θ and $\theta_{1}$. These constraints are realized in the polynomial form (Equation 65) through Equations (Equation 57)–(Equation 59) and (Equation 62), for the ranges of parameter values defined in Theorem 5.These conditions are not all satisfied by the polynomial
(73)$$\begin{array}{ccc} P_{3,r=2}\left(z\right) & = & z^{3}-2e^{i\pi }e^{-i\epsilon }z^{2}+2e^{i\pi }e^{-2i\epsilon }z-e^{-3i\epsilon }, \end{array}$$
which is of the form (Equation 52) with r=2, $\theta_{1}=-\epsilon +\pi$ and θ=2π−3ϵ. It is easy to see that this polynomial coincides with
(74)$$\begin{array}{ccc} P_{3,r=2}\left(z\right) & = & z^{3}-\left(1+3e^{i\pi }\right)e^{-i\epsilon }z^{2}+\left(1+3e^{-i\pi }\right)e^{-2i\epsilon }z-e^{-3i\epsilon }, \end{array}$$
which has the form (Equation 65) with θ=2π, except for the fact that $r_{2}\notin \left[0,2\right]$, as is instead required in Theorem 5 because of Theorem 2. As a consequence, the polynomial (Equation 74) or, equivalently, (Equation 73) cannot be the characteristic polynomial of a unitary 3×3 matrix.Polynomial (Equation 73) obviously admits an ACM of the form (Equation 20), which is obtained using the general protocol in Section 3. Moreover, the insertion of $r_{2}=3$ and θ=2π in matrix (Equation 67) leads to another ACM of polynomial (Equation 73) of the form
(75)$$W_{3nu}=\left(\begin{array}{ccc} -\frac{3}{2}e^{-i\epsilon }\; & -\frac{i}{2}\sqrt{5}e^{-i\epsilon }\; & 0\\ \frac{i}{2}\sqrt{5}e^{-i\epsilon }\; & -\frac{3}{2}e^{-i\epsilon }\; & 0\\ 0 & 0 & e^{-i\epsilon } \end{array}\right),$$
which is manifestly non-unitary. In general, $W_{3nu}$ and the ACM of the form (Equation 20) are not similar. A sufficient condition for their similarity is that their three common complex eigenvalues are distinct. In this case, in fact, both matrices are surely diagonalizable [2] and, therefore, traceable (in general, not unitarily) to the same diagonal matrix.

## 6. Discussion and Conclusions

Finding a matrix with an assigned characteristic polynomial is a classic inverse problem solved by Frobenius a long time ago. In Section 2 we observed that there are infinitely many other solutions, generally not equivalent to the one found by Frobenius. The exhaustive description of the set $S(P_{n}\left(z\right))$ of all complex matrices sharing the same characteristic polynomial is still an open problem. Among the reasons for the missing solution to this problem, it is useful to consider that, even if $S(P_{n}\left(z\right))$ is invariant under similarity transformations, it includes non-similar (and in general not even equivalent) matrices. Another problem is that the structure of the non-empty subset of non-sparse CMs belonging to $S(P_{n}\left(z\right))$, unlike the set of sparse CMs [35], has not yet been fully characterized.

A related inverse problem, stimulated by applications of current interest to both physicists and mathematicians, is the search for ACMs *constrained* to possess prescribed structural properties, such as, for example, unitarity, positive semi-definiteness, or Hermiticity.

The first focus of this study is the construction of a new ACM of a generic monic and complex third-degree polynomial $P_{3}\left(z\right)$, characterized by versatility for applications. This objective is pursued by parameterizing the elements of the ACM in such a way that they lend themselves to additional constraints dictated by specific problems (of which only the structural properties are exploited). To the best of our knowledge, our investigation of inverse problems of this kind opens a new fruitful chapter in this research area whose central goal is the proposal of new ACMs which, in particular, can be CMs. We address the three above-mentioned constrained inverse problems, providing methodology and results that aim for broad applicability and are potentially transferable to solving analogous problems involving higher-degree polynomials.

The adopted step-by-step approach builds new specific classes of unconstrained or constrained matrices as ACMs of suitably given polynomials, relaxing from the outset any condition on the degree of the minimal polynomial. In particular, the strategy implemented, as well as the mathematical tools used, is not influenced by any FCM-based or FCM-inspired technique.

The elements of the ACM that we construct as a solution to the general inverse problem are single-valued complex functions of the coefficients of the given generic polynomial. Exploiting the structural properties of this ACM, we find the algebraic expressions for the three, generally complex, roots of its complex characteristic polynomial.

It is remarkable that, when the polynomial becomes real, the associated general ACM smoothly becomes Hermitian under easy-to-find necessary and sufficient conditions on the coefficients of the polynomial described by (Equation 45). Using simply the fact that the ACM becomes Hermitian if and only if Δ≤0, we are able to extend the trigonometric representation of the three roots of the real polynomial to all possible cases, that is even when (Equation 29) does not hold. This representation is obtained without resorting to the well-known Cardano-Del Ferro formulas.

We emphasize that the FCM of a characteristic polynomial does not undergo any structural change when the coefficients of the complex polynomial become real, thus providing no additional information on the polynomial roots. For this reason, we claim that the FCM has a lower flexibility than our ACM. We show how to use this flexibility to obtain an ACM of a prescribed characteristic polynomial on demand, applying our procedure to the important problem of finding a density matrix, particularly that of a qutrit.

A second, novel constrained inverse problem addressed here consists in finding a unitary ACM of a generic polynomial with three roots of modulus one. Excluding the trivial case in which the unitary ACM can be directly given in diagonal form, we first reach the intermediate goal (interesting in itself) of parameterizing $c_{1}$, $c_{2}$, and $c_{3}$ in the polynomial (Equation 5) so as to set the necessary conditions on the structure of a polynomial to admit an ACM. By setting appropriate relations between the parameters through coupled geometric and analytical considerations, we further constrain the polynomial structure in such a way as to identify the set $\left[P_{3\epsilon }\left(z\right)\right]$ of all and only the third-degree polynomials that admit a unitary ACM. Then, we conclude our analysis by constructing the associated ACM.

The results of this study can be further explored and usefully applied to physical and mathematical contexts, including, at their intersection, the research area of quantum computing. For example, for time-dependent parameters, the prescription of a time-dependent characteristic polynomial of $\left[P_{3}\left(z\right)\right]$ or $\left[P_{3\epsilon }\left(z\right)\right]$ leads to a unitary time-dependent matrix, hence to the pertinent time-dependent Hamiltonian that generates the time evolution of a qutrit, whose properties can thus be traced back to the prescription of $P{}_{3}\left[z\right]$. In mathematical contexts, the analysis developed in this study can be extended to polynomials of a higher degree. For example, in the case of a fifth-degree polynomial, our protocol may provide conditions on the coefficients of the polynomial such that its roots are real. The rich analysis enabled by the use of third-degree polynomials in this study sets a clearer basis to conceive the extension of the analysis to higher-degree polynomials.

## Data Availability

All necessary data are contained in the article.

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
