# Peer review of "Hermitian and Unitary Almost-Companion Matrices of Polynomials on Demand"

_entropy, 2023, doi:10.3390/e25020309_

Round 1

Reviewer 1 Report

The authors present the concept of ACM and apply it, for instance, to cubic complex algebraic equations and for the construction of a qutrit.

The paper is interesting for an audience working in the specific fields, also because of possible applications in the context of quantum information.

I have a main comment related to the content of the paper.

*) CM's have been previously used in the context of graph states (see for instance Spengler and Kraus, Phys. Rev. A 88, 052323, 2013). Do the authors see some connections with their results? In may opinion, they should include some comments.

Author Response

We thank the Reviewer for bringing the interesting study of Spenser and Kraus to our attention. Our analysis is, indeed, not related to this study. Spenser and Kraus use a similarity transformation on a Frobenius companion matrix to obtain a symmetric matrix that provides a compact representation of a complete set of mutually unbiased bases.

We, instead, depart from the concept of companion matrix since the very beginning of our study, introducing the concept of “almost-companion matrix”. Furthermore, the matrices that we consider generally do not have integer elements, are not symmetric, and do not need to be related to any basis set. However, the relationship to a basis of physical states can be usefully set where the matrix is constructed to represent, for example, the density operator of a physical system. While our formalism may also find useful application in relation to mutually unbiased bases in future studies, exploring this fact is beyond the scope and conception of the present study.

Nevertheless, we found it useful to mention the paper of Spengler and Krauss in the general introduction of our revised manuscript (new reference [32]), where we added the sentence (please see page 3, line 83, of the revised manuscript)

“Furthermore, recent studies [32] have shown the opportunity of using CMs in mathematical constructions useful to the investigation of quantum entanglement, quantum state tomography, and quantum information in general.”

Reviewer 2 Report

Authors study relations of polynomials with matricies on the example of density matricies associated with quantum states and unitary matricies; this problem is interesting in physics, since the evolution of quantum states and symmetry properties of the states are provided.

The consideration is done in mathematical style with theorems. The paper is interesting

and, in principle, can be published  because the example of qutrit states is discussed.

Also, the generalization to other properties of the discussed polynomials and their relations

to different quasidistributions and probability distributions determining the qutrit states

can be considered. It is worthy to recommend the authors to mention how their method

depends on the representation of the matrices, which are interesting for applications in

quantum mechanics.

Author Response

We thank the Reviewer for their interesting comments.

Our construction of ACMs does not need the definition and support of any vectorial space. Then, if we want that our ACM represents, for example, the density operator of a qutrit, the link to the basis states of the physical system clearly appears. To stress this point, on page 11 or the revised manuscript we added the following sentence:

“It is worth noting that our approach itself does not require the support of a vector space, while the relationship to a basis of physical states appears in this application to a qutrit.”

Our analysis may open new scenarios in the approaches to describe the states of quantum systems, also where quasi-distributions and probability distributions are employed. However, the exploration of these interesting and broad scenarios goes beyond this first study, where ACMs are introduced and their properties are investigated.